# Probing the Nature of Chemical Bonds by Atomic Force Microscopy

**DOI:** 10.3390/molecules26134068

**Published:** 2021-07-03

**Authors:** Franz J. Giessibl

**Affiliations:** Chair for Quantum Nanoscience, Institute of Experimental and Applied Physics, University of Regensburg, D-93040 Regensburg, Germany; franz.giessibl@ur.de

**Keywords:** chemical bond, covalent bond, ionic bond, hydrogen bond, metallic bond, hybridization, atomic force microscopy

## Abstract

The nature of the chemical bond is important in all natural sciences, ranging from biology to chemistry, physics and materials science. The atomic force microscope (AFM) allows to put a single chemical bond on the test bench, probing its strength and angular dependence. We review experimental AFM data, covering precise studies of van-der-Waals-, covalent-, ionic-, metallic- and hydrogen bonds as well as bonds between artificial and natural atoms. Further, we discuss some of the density functional theory calculations that are related to the experimental studies of the chemical bonds. A description of frequency modulation AFM, the most precise AFM method, discusses some of the experimental challenges in measuring bonding forces. In frequency modulation AFM, forces between the tip of an oscillating cantilever change its frequency. Initially, cantilevers were made mainly from silicon. Most of the high precision measurements of bonding strengths by AFM became possible with a technology transfer from the quartz watch technology to AFM by using quartz-based cantilevers (“qPlus force sensors”), briefly described here.

## 1. Introduction

Linus Pauling was born in 28 February 1901 in Portland, Oregon (USA) and dedicated much of his life to the understanding of chemical bonds, leaving a landmark textbook on the nature of chemical bonds that is still very valuable today [1].

Pauling passed away on 19 August 1994 in Big Sur, California (USA). At this time, the strong interaction offered by covalent chemical bonds was utilized only 100 miles away at Park Scientific Instruments in Sunnyvale, a manufacturer of atomic force microscopes (AFM) [2], for a first demonstration that AFM could obtain atomic resolution in vacuum. Atomic resolution of the enigmatic Si(111)–(7×7) surface by scanning tunneling microscopy (STM) [3,4] convinced the scientific community that STM really resolves atoms, therefore the achievement of the same feat by AFM in a paper that was submitted to *Science* eleven days after Paulings death [5] was an important step. AFM relies on the force interaction of a sharp tip with the surface of a sample. Even when working in vacuum, the forces that act between tip and sample can be complex, as they have various origins. Van-der-Waals forces are always there, covalent bonding (directional and nondirectional), ionic bonding, hydrogen bonding and Pauli repulsion can all be a factor. AFM not only allows us to image samples using those forces, but it also enables a direct measurement of bonding forces as a function of distance—*an interaction of orbitals through space and through bonds* as phrased by R. Hoffmann [6]. Experimental examples of studying these various types of bonds by AFM will be covered in this short review.

When performing AFM in ultrahigh vacuum (UHV) at low temperatures, very low noise measurements become possible that essentially allow one to put a single chemical bond on a test bench. For an overview on high resolution AFM in vacuum, see review articles [7,8,9,10] and books [11,12,13,14], for biological environments see [15,16].

Whenever a new scientific tool such as a powerful microscope is developed, controversies can arise in early explorations of new research areas. One example was the observation of hydrogen bonds, covered in some detail in Section 3.6. Another example of initial controversy is the observation of a strong angular dependence of covalent bonds that result in subatomic contrasts in the interaction between tip and sample. First results were claimed in 2000 [17], although it stirred up some controversies [18], giving rise to fruitful further developments as discussed in Section 3.9.

The goal of this paper is to report on the possibilities to precisely study chemical bonds by AFM, concluding that all types of chemical bonds are open to investigation by this powerful tool.

## 2. Materials and Methods

We use atomic force microscopy, introduced in 1986 by Binnig, Quate and Gerber [2] that probes the interaction between a sharp tip and a surface. Typically, experiments are performed in UHV and often even at liquid helium temperatures of about 4 K.

AFM faces four challenges inherent to the physics of the tip-sample interaction: instability when approaching a soft cantilever to a surface, the concealment of chemical bonding forces by much larger long-range van-der-Waals forces, noise in measuring small forces and a nonmonotonic imaging signal, explained in detail in section III of [8]. Frequency modulation (FM–) AFM [19,20] allows one to overcome the first three of these challenges if small amplitudes and stiff cantilevers are used [8].

In FM-AFM, a cantilever is driven by applying positive feedback to oscillate at frequency *f* at a constant amplitude *A* as shown in Figure 1. The frequency *f* is given by the sum of the eigenfrequency of the free cantilever $f_{0}$ and the frequency shift Δf, where $\Delta f=f-f_{0}$ is given by
(1)$$\Delta f\left(z\right)=\frac{f_{0}}{2k}k_{ts}\left(z\right)$$
as introduced in [19,20]. However, although this formula is only correct if $k_{ts}\left(z\right)$ is constant over the z−range covered by the oscillating cantilever.

If $k_{ts}\left(z\right)$ varies significantly over the *z*-range covered by the cantilever, as it does in most practical cases, we find
(2)$$\Delta f\left(z\right)=\frac{f_{0}}{\pi k}\int_{-1}^{1}k_{ts}\left(z-\zeta A\right)\sqrt{1-\zeta^{2}}d\zeta$$
as derived in [21] using the Hamilton-Jacobi mechanism, while the descriptive form of the equation as a convolution of the tip-sample force gradient with a semispherical weight function was introduced in [22]. Here, we use a coordinate system where x,y denote the sample plane and *z* is normal to the sample.

The origin of the frequency shift is the gradient of the tip-sample force $F_{ts}$ given by
(3)$$k_{ts}\left(z\right)=-\frac{\partial F_{ts}}{\partial z}$$
where the force is the gradient of the tip-sample potential $V_{ts}$ given by
(4)$$F_{ts}\left(z\right)=-\frac{\partial V_{ts}}{\partial z}$$

A major complication of AFM versus scanning tunneling microscopy (STM) [3,23,24], its atomically resolving predecessor, is that the chemical bonding forces $F_{sr}$ with a short range at decay length $\lambda_{sr}$ that allow in principle obtaining atomic resolution are concealed behind van-der-Waals forces $F_{vdW}$ that are larger in magnitude and also have a longer range with a decay length $\lambda_{vdW}\gg \lambda_{sr}$. Equation (Equation 2) insinuates that FM-AFM is sensitive to the force gradient F/λ, leading to a natural suppression of long-range forces. However, analysis of Equation (Equation 2) shows that for oscillation amplitudes A≫λ, the frequency shift is proportional to $F\sqrt{\lambda }$ [21], emphasizing the contribution of the unwanted long-range forces.

Let us look at a realistic example of tip-sample interactions. One type of forces that are ever present are van-der-Waals (vdW) forces [25], an induced dipole-dipole interaction with its characteristic $1/r^{6}$ potential for two atoms interacting with one another for distances up to about 100 nm:(5)$$V_{atom-atom\;vdW}\left(r\right)=-\frac{C_{vdW}}{r^{6}}$$
where $C_{vdW}$, sometimes also called $C_{6}$, is the atomic vdW constant. For larger distances, the phase differences between the induced dipole moments become important, leading to a retarded interaction that is not of practical interest here [25]. The vdW interaction is additive, thus two mesoscopic bodies interact depending on their shape, where the Hamaker constant $A_{H}$ given by
(6)$$A_{H}=\pi^{2}C_{vdW}\rho_{t}\rho_{s}$$
becomes important. The items $\rho_{t}$ and $\rho_{s}$ are the atom densities (number of atoms per volume, unit 1/m${}^{3}$) of the two materials making up tip (t) and sample (s). For many combinations of solids, $A_{H}\approx 1$ eV.

The interaction law between two macroscopic bodies depends on their shape. For a sphere next to a flat at distance *z*, the force is proportional to $1/z^{2}$ [25] and for a conical tip next to a flat sample, we find
(7)$$F_{vdW\;conical}=-A_{H}\frac{\tan^{2}\left(\alpha /2\right)}{6(z+z_{\mathrm{offset}})}$$
by performing the corresponding Hamaker integration [21]. The full opening angle of the cone is α and $z_{\mathrm{offset}}$ is a small displacement that takes care of the properties of good tips that often have a small cluster (or CO molecule) protruding at the tip apex.

The potential energy of chemical bonds can often be modelled by a Morse potential
(8)$$V_{Morse}\left(r\right)=E_{bond}\left[-2\exp\left(-\left(r-r_{0}\right)/\lambda \right)+\exp\left(-2\left(r-r_{0}\right)/\lambda \right)\right]$$
where $E_{bond}$ is the bond energy, $r_{0}$ is the equilibrium distance and λ is the decay length.

When $r>r_{0}$, we can estimate the attractive part of the bond by a short-range force *F* with
(9)$$F_{sr}\left(r\right)=F_{0}\exp\left(-r/\lambda \right).$$

We can now demonstrate the effect of amplitude *A* on the contribution of long- and short-range forces to frequency shifts in a realistic Gedanken experiment. Here, we assume that our cantilever has a stiffness of k=1800 N/m and an eigenfrequency of $f_{0}=30$ kHz. The tip-sample interaction is comprised of a long-range vdW contribution due to a conical tip with an opening angle of $\alpha =150^{\circ }$, a Hamaker constant of $A_{H}=1$ eV, a tip offset of $z_{\mathrm{offset}}=500$ pm and a short range force of $F_{0}=-500$ pN with λ=50 pm. This choice of parameters results in a long range attractive force of −744 pN, i.e., 50% larger than the short-range force at the closest point during the oscillation cycle of the tip.

The frequency shift that emerges from such an interaction as a function of amplitude is shown in Figure 2. The inset shows the tip model with the conical tip next to a flat sample. The red line marks the short-range contribution to the frequency shift. Although the short range force is weaker than the long-range force, its contribution to the frequency shift for A=1 pm is about −82 Hz, while the contribution of the long-range force is only −12 Hz. At an amplitude of A=100 pm, the short-range force contributes about −18 Hz, while the contribution of the long-range force is only −8.8 Hz and at A=10 nm, the long-range force clearly dominates with −140 mHz, while the short-range force only contributes about −24 mHz at this amplitude.

The result demonstrates that small amplitudes should be used to maximize the contribution of short-range forces. The cost of reducing the amplitude, however, is increased noise in the frequency measurement, as frequency noise is proportional to 1/A [8]. An optimal trade off between reduced frequency shifts (signal) and noise in the frequency shift, i.e., an optimal signal-to-noise ratio, is obtained for an amplitude of
(10)$$A_{\mathrm{optimal}}\approx 1.55\lambda$$
as derived in [8,26]. Comparison with Figure 2 shows, that $A_{\mathrm{optimal}}\approx 75$ pm with λ=50 pm still yields a sizable suppression of long-range forces.

The additional requirement of a stable oscillation in the presence of strong attractive forces requires a sufficient stiffness of the cantilever. One embodiment of a stiff cantilever, that has the additional benefits of a high frequency stability and electrical self sensing (rather than requiring the complexity of optical deflection detection) is the so called qPlus sensor [27,28,29] shown in Figure 3. The sensor generates an alternating current when it oscillates that is amplified by a current-to-voltage converter [30].

Today, this sensor is used in most low temperature AFMs and most of the data shown in this review was obtained with it.

To convert experimental frequency shifts to forces for comparison with theory, an inverse problem must be solved, where the Sader-Jarvis formalism [31] or the Matrix inversion [22] can be used. It is noted, that both inversion algorithms can be ill-posed for complicated force curves with one or even more inflection points and restrictions on the proper choice of oscillation amplitudes *A* apply [32,33,34].

## 3. Results

### 3.1. Covalent Bonds

In the year 1995, the first result of an atomically resolved surface using FM-AFM, the Si(111)–(7×7) surface [5], was published. The experiment utilized a silicon cantilever with a stiffness of k=17 N/m at an amplitude of A=34 nm, clearly outside of what was found above as optimal. Nevertheless, atomic resolution was obtained because covalent bonds can be quite strong, on the order of 2 nN for two Si atoms interacting via covalent bonds as shown in Figure 4. The image was recorded with a scanning speed of 3.2 horizontal lines per second starting from the bottom to the top, so it took 1 min and 20 s to record the whole image. In the lower image section, atomic contrast is visible but weak. In the upper third of the image, strong contrast is shown but lost again due to the change of the tips front section, indicating strong interaction between tip and sample.

The covalent nature of these bonds was identified by density functional theory (DFT) calculations by Perez et al. [35]. Figure 5A,B adapted from [35,36], show the formation of a covalent bond between the tip and the adatoms of Si(111)–(5×5), the smaller unit cell was chosen to keep the computational effort reasonable. The charge density plots in Figure 5 show how the covalent bond manifests itself by a build up of charge between the ligands in a similar way as in bonds in bulk group IV semiconductors [37].

Figure 6, again from Perez et al. [36] shows that the maximal attractive force of the covalent bond between a silicon tip and an adatom of silicon reaches about 2.5 nN.

Precise force spectroscopy measurements by Lantz et al. [38] displayed in Figure 7 confirmed the DFT results. Lantz et al. used a silicon cantilever (Nanosensors) with a stiffness of k=48 N/m, an amplitude of A=6.1 nm and an eigenfrequency of $f_{0}\;=\;172,509.55$ Hz.

Lantz et al. took great care to preserve the sharpness of the tip to obtain only weak vdW forces as evident from the small difference between the total force (red curve in Figure 7) and short-range force curves (yellow in Figure 7). The Si tip was initially covered by a native oxide that only provided weak atomic contrast and weak forces. Gently picking up a few Si atoms from the surface allowed to obtain strong reactivity and thus strong short-range forces.

Sugimoto et al. measured the maximal forces that covalent bonds between Si-Si, Si-Sn and Si-Pb atoms can exert in a refined experiment at room temperature using Si cantilevers [39]. To ensure that the long-range force is similar for these three combinations, they prepared a Si(111) surface where some of the Si adatoms were replaced by other group IV elements as Sn and Pb. The three different chemical elements showed different magnitudes of short-range forces, enabling to identify the chemical element within these three candidates (Si, Sn and Pb) by precision force spectroscopy [39].

### 3.2. Van-der-Waals Bonds

Van-der-Waals forces of mesoscopic tips are essentially of a long-range nature with a 1/z distance dependence for conical tips as noted in Equation (Equation 7).

In principle, atomic vdW forces are short-range with their distance dependence of $1/z^{7}$ (see Equation (Equation 5)). Kawai et al. [40] found a smart experimental situation where the atomic contribution of vdW forces could be extracted as explained below. They used a low-temperature AFM with a qPlus sensor that had a Xe atom adsorbed on its metallic tip to probe its interaction with Ar, Kr and Xe atoms. The noble gas atoms were confined into a two-dimensional metal–organic framework as an anchor net for the Ar, Kr and Xe gas atoms. The two-dimensional framework was created by depositing the perylene derivative 4,9-diaminoperylene-quinone-3,10-diimine (DPDI) on a clean Cu(111) surface and annealing at 300 ${}^{\circ }$C [40]. Taking the difference of force spectra aquired over an empty node in the framework and over a node filled with a noble gas atom allowed to extract the atom-atom interactions as illustrated in Figure 8a,b.

Kawai et al. found an interaction energy of 18, 26 and 36 meV for Xe-Ar, Xe-Kr and Xe-Xe junctions, respectively, as displayed in Figure 8. The energy curves were integrated from the experimental frequency shift data and the atom-atom interactions were extracted by subtracting the background forces as explained in Figure 8a,b and [40].

### 3.3. Metallic Bonds

The ability to move single atoms where one wants them as demonstrated by Eigler and Schweizer in 1990 [41] has opened a new capability to mankind. In this first example, Xe atoms adsorbed on a Ni(110) surface were manipulated with the tip of a scanning tunneling microscope. The driving force behind this atomic manipulation was the tunable bond between the metallic STM tip and the Xe surface atoms. A few years later, Crommie, Lutz and Eigler [42] manipulated Fe adatoms on a Cu(111) surface to build a quantum corral, again using a bond between two metal atoms—the front atom of the STM tip and the Fe adatom on the Cu surface where the force was tuned by adjusting the distance of the STM tip to the surface similar as in the Xe example.

About a decade later, I had the opportunity to work in the IBM Almaden research laboratory where the pioneering experiments of Eigler et al. were conducted. Together with Andreas Heinrich, Chris Lutz, Cyrus Hirjibehedin, Markus Ternes and Bruce Mellior, we built a qPlus sensor into the original Eigler low-temperature STM, rendering it into a combined STM and AFM. With this instrument, the force that acts when a single atom is moved was measured for the first time [43,44]. Reversible bond formation between a gold atom and a pentacene molecule had been demonstrated before by STM [45], and the capablity to measure the forces by AFM during such processes was a welcome addition. A few more years later, Ternes et al. measured the chemical bonding force between two metal atoms, supplemented by DFT calculations by Jelinek et al. [46], see Figure 9. The separation between long- and short-range forces was achieved by taking the difference between a force versus distance spectrum over a flat surface and an adatom (see inset of Figure 9a). The bonding force amounted to about 2 nN as shown in Figure 9c. In the high precision force spectroscopy work of Lantz et al. [38], the short range force contribution was extracted by taking the difference between forces over an adatom and the large cornerhole.

### 3.4. Ionic Bonds

Conceptually, ionic bonds are probably the simplest to understand as the electrostatic interaction energy is straightforward with Equation (Equation 11):(11)$$V_{atom-atom\;est}\left(r\right)=\frac{1}{4\pi \epsilon_{0}}\frac{q_{1}q_{2}}{r},$$
where $\epsilon_{0}$ is the dielectric constant and $q_{1}$ and $q_{2}$ are the charges that interact. One caveat is that the simple electrostatic equation only addresses the interaction satisfactorly if the atoms are far enough apart, i.e., if electronic overlap can be neglected.

Ionic crystals of the rocksalt structure such as NaCl, KBr etc. are easy to prepare in an UHV environment by cleaving, yielding high quality (001) faces with little defects or steps. However, it is nontrivial to assess whether a specific atomic site on the surface is an anion or a cation—a positively charged tip will attract the anion and repel the cation, while the opposite holds for a negatively charged tip.

Foster et al. [47] suggested to use CaF${}_{2}$, as its natural cleavage surface plane (111) exposes only F${}^{-}$ ions at the top, with a layer of Ca${}^{++}$ ions underneath and another F${}^{-}$ layer below (see Figure 10A,B).

As already found by Lennard-Jones and Dent in 1928 [48], the electrostatic field of an ionic crystal decays exponentially normal to the surface although the Coulomb interaction energy decays at the inverse distance. For the lattice parameters of CaF${}_{2}$, a vertical decay length of λ=53.2 pm results [49]. In theory, for every distance increase of λ·ln(10)=122.3 pm, the electric field decays by a factor of 10 as shown in Figure 10C. Experimentally, this exponential law was confirmed with great precision over three orders of magnitude ranging from a contrast of 100 fN to 40 pN, spanning a vertical distance of more than 300 pm in Figure 2 of [49].

The single point charge tip model according to Figure 10C can be refined by utilizing the so called COFI (carbon monoxide front atom identification) method introduced by Welker et al. [51] to a tip composed of multiple point charges. In COFI, a CO molecule adsorbed vertically on a metal surface such as Cu(111) or Pt(111) [52] is used to probe the front section of the tip. The COFI technique allows the creation of an image of the front section of the tip, enabling one to refine the expected image that such a non-ideal tip creates.

Figure 11A shows the COFI portrait of the tip that is modelled by one main tip and three minitips surrounding it. The calculated image of the CaF${}_{2}$(111) surface based on the tip structure deduced from COFI is displayed in Figure 11B. Although the calculated image looks as if it fully reflects the threefold symmetry of the underlying crystal lattice, analysis of the red, green and blue traces in Figure 11B,C shows deviations that can be fully explained by taking the contributions of the main tip and the three minitips into account. The dashed black traces in Figure 11C are the theoretical curves for a single charge tip. The colored dots are experimental data and the solid colored curves show the simulated images taking all four tips into account. The match is very convincing, with a 99.5% agreement between theory based solely on Coulomb interaction and experiment as explained in detail in [49].

### 3.5. Antibonds Due to Pauli Repulsion and Their Role in Imaging Organic Molecules

The usual way to reach atomic resolution by AFM is to work in a noncontact mode employing attractive chemical bonding forces. However, using chemically inert tips allows one to probe the repulsive regime without moving weakly bonded molecules on a surface or even damaging the sample.

The most widely used tip that accomplishes this is a metal tip that is terminated by a single CO molecule. The CO molecule bonds to the metal end of the tip similar to a metal carbonyl, i.e., with the C atom sticking to the metal and the O atom pointing into the vacuum as a chemically inert tip termination as first found in STM experiments by Bartels et al. [53]. Gross et al. discovered the great utility of CO terminated tips in a pioneering experiment on atomic imaging of pentacene [54] displayed in Figure 12. Although other inert tip terminations such as noble gases [55] and Cu oxide tips [56,57,58] have been studied, CO tip terminations are still popular in spite of their lateral softness [59] as emphasized in a recent review by Gross et al. [60].

CO terminated tips also enable atomic resolution of metal clusters [61]. In spite of their lateral flexibility [59], these tips have the capability to atomically manipulate metal atoms on surfaces [62] and to study the chemical reactivity of small metal clusters as a function of the atomic site [63]. Surprisingly, the reactivity of CO terminated tips scatters somewhat even if the CO tip termination sits at the foremost single front atom of the tip, apparently depending on the tip cluster behind the metal front atom that holds the CO tip molecule (see Figure 1 in [63]).

When imaging organic molecules, the atomic contrast formation is based on Pauli repulsion as calculated by Moll et al. [64]. Figure 13 shows how the frequency shift image looks as a function of distance. When approaching the tip, one first sees a bathtub-like field of vdW attraction. Upon further approach, the bonds and hexagons of pentacene start to appear in the calculated image of Figure 13. As it is required to strike a fine balance between vdW attraction and Pauli repulsion, the imaging of molecules that extend into the third dimension such as dibenzo[a,h]thianthrene is possible but challenging [65].

The utility of CO terminated tips even extends to very tiny molecules such as water, where a pentagonal chain of water molcules on Cu(110) [66] and interfacial water on NaCl [67,68] was observed.

The calculations by Moll et al. [64] showed that for pentacene, the interpretation that CO terminated tips probe the total charge density is quite accurate. An important question is, whether CO terminated tips always interact mainly by Pauli repulsion, aside from weak vdW attraction before contact.

Figure 13 shows that indeed, contrast formation is due to a strong onset of Pauli repulsion, leading to drastic changes when reducing the height for merely 20 pm from the frame at 4.95 Å to 4.75 Å.

Of course, one needs to be cautious with any generalizations. For example, it was held as a truth for a long time that noble gases are totally inert, while Linus Pauling foresaw that Xe would be able to bond to Fl [69] in a prediction that needed three decades to be verified experimentally [70].

Huber et al. [71] found that CO tips that interact with Cu or Fe atoms can indeed form bonds once a repulsive Pauli barrier is penetrated. This issue will be addressed in detail in Section 3.7.

### 3.6. Hydrogen Bonds

Hydrogen bonds are caused by electrostatic interaction between a positively charged H atom and a negatively charged ligand. Probing such a bond with a CO terminated tip could cause a lateral force if one assumes that the CO terminated tip has a negative charge at its very end as evident from its interaction with a Cu${}_{2}$N surface [72], Cl vacancies [73] and a CaF${}_{2}$ surface [74]. If the hydrogen bond leads to an increased charge density between the ligands, a weak signature of repulsion might emerge.

Zhang et al. [75] found lines in AFM images recorded with CO terminated tips between 8-hydroxyquinoline (hq) molecules assembled on a Cu(111) surface at locations where hydrogen bonds are expected as displayed in Figure 14.

Hämäläinen et al. [76] performed AFM experiments with CO terminated tips on an assembly of Bis(parar-pyridyl)acetylene (BPPA) molecules shown in Figure 15 and also found weak lines at locations where hydrogen bonds are expected, but a much stronger line (red arrow in Figure 15) where two nitrogen atoms are close and no hydrogen bond is expected. While this finding does not rule out that hydrogen bonds do show up as lines, these authors stressed that CO bending can cause artifacts that can be modelled with the so called probe-particle model after Hapala et al. [77]. Ellner et al. provided a theoretical study about the visibility of hydrogen bonds by AFM with CO terminated tips, concluding that the hydrogen bonds do not cause significant charge redistributions [78].

Kawai et al. [79] chose a different route to probe hydrogen bonds. They prepared three-dimensional hydrocarbons based on propellane derivatives that expose hydrogen atoms at the top and probed them with an Ag tip terminated with a CO molecule as shown in Figure 16. Two types of molecules, trinaphtho[3.3.3]propellane (TNP) and trifluorantheno[3.3.3]propellane (TFAP) were probed and the hydrogen bond was measured.

Recently, Wagner et al. [80] even probed four different hydroxyls on an In${}_{2}$O${}_{3}$(111) surface and found that the bonding strength of the hydrogen bonds allows to distinguish the proton affinity of these sites as displayed in Figure 17.

The question of the detectability of hydrogen bonds has also been addressed by A. Extance in [81].

### 3.7. Transition from Physisorption to Chemisorption

In 1932, Lennard-Jones suggested that molecules that adsorb on surfaces can find two different equilibrium distances—a weak adsorption due to vdW forces that can merge into a stronger chemical bond at closer distance once a repulsive barrier is overcome [82]. This transition encompasses a change from a vdW bond to a much stronger bond.

In 2019, Huber et al. [71] directly measured such a transition with an AFM with a CO terminated tip for certain surface atoms. Figure 18 shows the forces between a CO terminated tip and a silicon adatom on Cu(111) in the left column, a copper atom in the center column and an iron atom in the right column, reprinted from Figure S11 in [71] (a version in color is printed as Figure 2 in [71]). The first row shows the color coded vertical force along the vertical *z*-direction and the lateral *x*-direction. When approaching the Si adatom at the center in Figure 18A, a dark region appears that corresponds to vdW attraction, followed by Pauli repulsion for smaller distances. Repeating this experiment on top of a Cu adatom in Figure 18D leads to a completely different result. Again, initially vdW attraction is observed, followed by a reduction in the attractive force and a strong increase of attraction for even smaller distances. This behaviour only shows in the center of the Cu adatom at x≈0, while Pauli repulsion occurs at x≈±200 pm. For the Fe adatom in Figure 18G, a weak vdW attraction is followed by repulsion before strong attraction is observed.

The second row shows a top view at constant heights *z* indicated in the left bottom of Figure 18B,E,H. For the Si adatom in Figure 18B, Pauli repulsion is strongest in the center, while Cu and Fe show a repulsive ring with an attractive center in Figure 18E,H. The repulsive ring around the Fe adatom even displays three local maxima that are located over the hollow sites of the Cu(111) substrate.

The origin of this interesting contrast is revealed by DFT calculations displayed in the bottom row of Figure 18 and in Figure 19. The calculations in the bottom row of Figure 18 confirm the experimental constant height images in the second row.

The first column in Figure 19 shows the calculated force versus distance over the centers of the adatoms, while the second, third and forth column show the differential charge density plots.

The Si adatom interacts initially by vdW attraction, followed by Pauli repulsion without much change in the charge density plots.

The Cu and Fe adatoms in the second and third row show a significant charge rearrangement, indicative of strong directional bonding at close distance and confirming the transition from physisorption to chemisorption, i.e., a transition from vdW bonding, i.e. physical bonding, to significantly stronger chemical bonding.

### 3.8. Measuring the Very Weak Bond to an Artificial Atom with a Very Low Electron Density

In a natural atom, a fixed number of electrons is bound to a nucleus, giving rise to quantized energy states. Artificial atoms are arrangements of a fixed number of electrons that are also confined to a small space such as in quantum dots [83].

The quantum corral in its original form as introduced in 1993 by Crommie, Lutz and Eigler [42] is a circle of 48 Fe atoms with a radius of 7.13 nm on a Cu(111) surface. This corral holds electrons in 28 discrete energy levels with three quantum numbers, the number of radial nodes *n*, angular momentum *l* and magnetic quantum number [42], leading to wave functions $\psi_{n,l}\left(r\right)$. A total of 102 electrons are held in the corral, 10 of them are energetically close to the Fermi energy and thus visible to STM, 92 can only be observed by AFM [84].

Interestingly, the electronic shells of this artificial atom interact in a similar manner with metallic- and CO terminated AFM tips as natural atoms. The first two rows in Figure 20 show constant-height data recorded inside the 14.26 nm diameter corral with a size of 7×7 nm${}^{2}$, using a metal tip in the left column and a CO terminated tip in the right column. The corral states interact attractively with a metal tip and repulsively with a CO terminated tip. The data in the second row have been low pass filtered to suppress noise and the surface lattice image in the case of the CO tip.

The $\psi_{5,0}\left(r\right)$ state is very close to the Fermi energy. As the corral states are coupled to the bulk below the Cu(111) surface, the occupation of this state can vary with the interaction with the AFM tip. Attractive interaction will lower the energy of a state, repulsion will do the opposite.

The contrast of the radial ripples of the states is very low, for the metal tip, it ranges from about 1 pN to 50 fN, for the CO terminated tip it is even lower ranging from 200 fN to 20 fN as shown in Figure 20E,J.

### 3.9. Resolving the Directionality of Covalent Bonds by AFM—Subatomic Spatial Resolution

Linus Pauling would probably not have been surprised to learn that AFM can detect the directionality of bonds once the instrument provides sufficient force resolution, as chapter 4 of his book on the nature of chemical bonds [1] covers their directionality.

In 2000, our group published a report about the observation of subatomic spatial resolution by AFM [17], claiming that two dangling bonds in a Si atom were imaged (see Figure 21).

The explanation of the data is presented in Figure 22, where a special tip configuration was modelled using the Stillinger-Weber model for covalent bonds in silicon [85].

The AFM community was quite surprised by this result. A technical comment to our paper [18] suggested that using the Stillinger-Weber potential to explain the data was not state-of-the-art—DFT calculations would be more convincing. In addition, it was proposed that feedback oscillations might cause such images as well as the two orbital lobes were perpendicular to the fast scan direction. We explained that our experiment was performed under stable feedback conditions in our reply [18], and the DFT calculations of this experimental situation were performed a few years later by the group of Feng Liu [86]. In hindsight, we are grateful to this technical comment because it inspired us to design new experiments that address the directionality of chemical bonds, a field that is not yet very active, in spite of the exciting insights into the nature of chemical bonds it provides.

On a side note, Figure 21 was reprinted in a newspaper article about [17] in *Frankfurter Allgemeine Zeitung* in July 2000 [87], where it caught the eye of Gerhard Richter, one of the most influential living visual artists. The fuzzy image became the basis for Richters work “Erster Blick” [88], leading to inspiring exchanges of ideas and some more projects between Richter and the author over the following two decades.

Huang, Cuma and Liu [86] performed extensive first-principles calculations to simulate the AFM images of an adatom on the Si(111)–(7×7) surface imaged by a Si tip, studying a (001) and (111) orientation of the tip in Figure 23.

The DFT calculations confirmed the key result obtained by the initial modelling after the Stillinger-Weber potential: a (001) oriented Si tip would show a single maximum over the adatom for a medium distance, splitting into two features at close distance as shown in the left column of Figure 23, although the absolute heights were the splitting occurs were different by a fraction of an Å.

Admittedly, the initial experiment that demonstrated subatomic resolution still had drawbacks. First, it was created by the sample imaging the tip, while a microscope should have a well defined probe that images an unknown sample. Second, the tip that was responsible for the subatomic resolution had a random orientation of the orbital lobes. In later results that confirmed the work from 2000 [17], the two orbital lobes were rotated at an odd angle with respect to the fast scan direction (Figure 14 in [89]). The capability to prepare the tip to display a predetermined orientation of the lobes was not given yet. Other examples where the sample imaged subatomic structures followed: a Si(111)–(7×7) sample supposedly probing 4f electrons in samarium [90] and a carbon sample atom that presumably probed the subatomic structure of 3d electrons in tungsten [91] as confirmed with DFT by Wright and Solares [92].

The introduction of CO terminated probe tips in AFM by Gross et al. in 2009 [54] changed the situation, because it allowed the preparation of well defined probe tips that could be unleashed to probe new territory. Although metal tips that were first characterized by COFI [51] to display only a single metal front atom before picking up the CO termination still show some variety in their force curves to a specified sample situation [63], the CO terminated tips provide unprecedented reproducibility.

Figure 24 from [71] shows AFM data in the top row and STM data in the bottom row where a CO terminated tip probed an iron adatom. Figure 24B shows the registry of the three bumps on the repulsive ring with the Cu(111) lattice underneath.

We have reproduced the toroidal-shaped image of Fe with three local maxima shown in Figure 24A–C many times on different microscopes, demonstrating the achievement of subatomic spatial resolution under perfectly controlled conditions, a goal that kept us busy for 20 years.

## 4. Discussion

This short review has shown that AFM provides enlightening insights into the nature of chemical bonds, covering vdW, covalent, ionic, metallic and hydrogen bonds and demonstrating the strong angular dependence of covalent bonds. The field is still very active, with exciting new papers coming out every month. I imagine that the precise analysis of the strength and angular dependence of chemical bonding forces will provide fascinating insights into the nature of the chemical bond in the future.

Allmost all the data discussed here was recorded in UHV at low temperatures. Many chemical reactions and biological processes happen in liquid or in ambient conditions. Frequency modulation AFM also provides atomic resolution in ambient and liquid environments, both by using traditional silicon cantilevers [93,94] and qPlus sensors [95,96]. One exciting prospect of AFM in liquid is the combination with electrochemical STM [97], as insulated STM tips for electrochemical STM can easily be mounted on qPlus sensors.

## Figures and Tables

**Figure 1 molecules-26-04068-f001:**
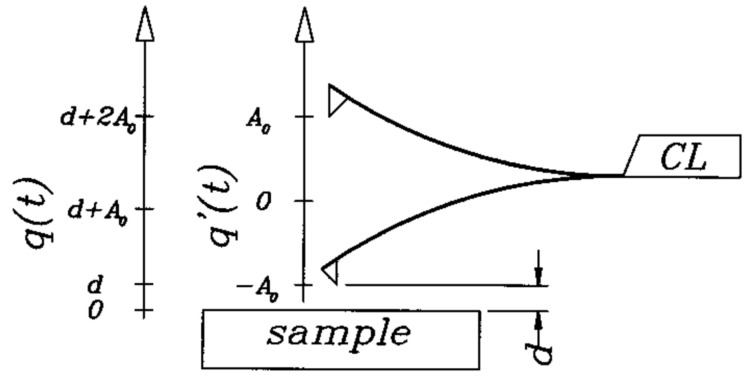
Schematic view of a vibrating cantilever close to a sample, it oscillates with an amplitude $A_{0}$, the closest *z*-distance is *d*. Soft cantilevers with a low spring constant *k* require a large oscillation amplitude to secure a stable oscillation. Adapted with permission by American Physical Society from Figure 1 in [21].

**Figure 2 molecules-26-04068-f002:**
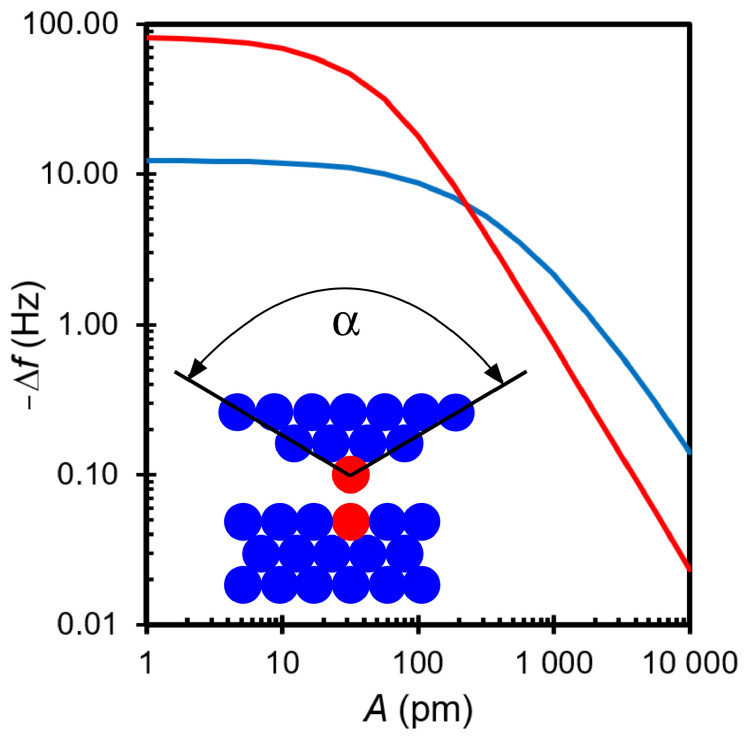
Frequency shift as a function of amplitude *A* for tip sample forces that are comprised of a van-der-Waals long range interaction and an exponential short range interaction.

**Figure 3 molecules-26-04068-f003:**
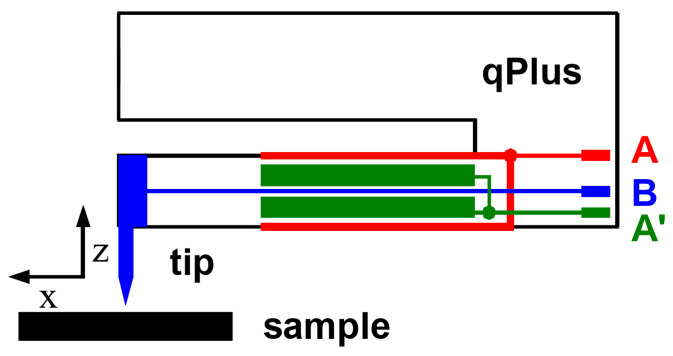
Schematic view of a qPlus force sensor with a tip mounted to it. The qPlus sensor is a cantilever made of a piezoelectric material (typically quartz). Originally, it was built from a quartz tuning fork taken from a Swatch electronic watch [27], but modern versions [29] have little in common with tuning forks and look as shown in the figure. The piezoelectric effect generates opposite electric charges at terminals **A** and **A${}^{\prime }$** when the beam is deflected, yielding an ac-current when the beam oscillates. If a metallic tip is used, STM and AFM can be performed in parallel and electrode **B** connects to a tunneling current amplifier. For detailed explanations and photographs of these sensors, see section III and Figures 9–13 in [29].

**Figure 4 molecules-26-04068-f004:**
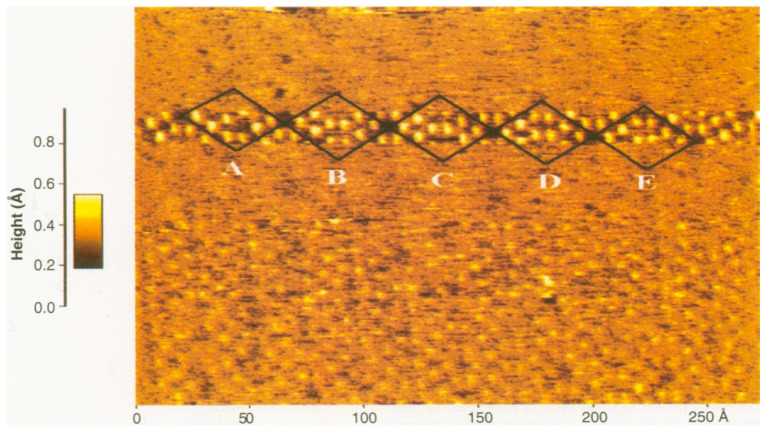
First atomically resolved AFM image of Si(111)–(7×7). The image was recorded with a Si cantilever with an eigenfrequency $f_{0}=114,224$ Hz at an oscillation amplitude of A=34 nm in the topographic mode with a constant frequency shift of Δf=−70 Hz. Adapted with permission by the American Association for the Advancement of Science from Figure 2 in [5].

**Figure 5 molecules-26-04068-f005:**
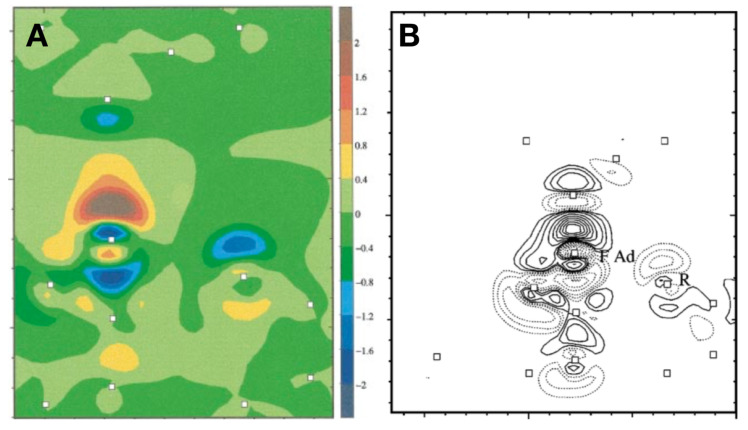
DFT calculations of covalent bond formation between a Si tip and Si surface atoms. (**A**) Charge density difference between the self- consistent electron density of the interacting system and the superposition of the densities of the isolated tip and the clean surface. The plot corresponds to a plane perpendicular to the surface along the long diagonal (only part of the unfaulted half of the unit cell is shown). The tip is on top of the adatom on the long diagonal in the faulted half of the unit cell (the point of maximum bonding energy and normal force in Figure 1). The squares indicate the position of the atoms in that plane both in the tip and in the Si surface. The contours are in units of $10^{-2}$ electrons/Å${}^{3}$. Notice the transfer of charge to the adatom dangling bond from the backbond and the dangling bond in the rest atom. (**B**) Charge-density difference between the self-consistent electronic density of the interacting system and the superposition of the densities of the isolated tip and surface. The plot is taken on a plane perpendicular to the surface along the long diagonal in the faulted half of the unit cell. The tip is on top of the adatom in the faulted half. The tip-surface distance is 3.5 Å. The squares indicate the position of the Si atoms in that plane both in the tip and in the Si surface. The solid (dotted) lines indicate an increase (decrease) of electronic charge; the contours correspond to ±0.4, ±1, ±2 (as in Figure 2 in reference [35]), ±4, ±6,±8, ±10, ±12, ±14 and $\pm 16\times 10^{-2}$ electrons/Å${}^{3}$. Notice the transfer of charge to the adatom dangling bond from the backbond and the dangling bond in the restatom in the faulted half. Adapted with permission from American Physical Society from Figure 2 in [35] (A) and Figure 6 in [36] (B).

**Figure 6 molecules-26-04068-f006:**
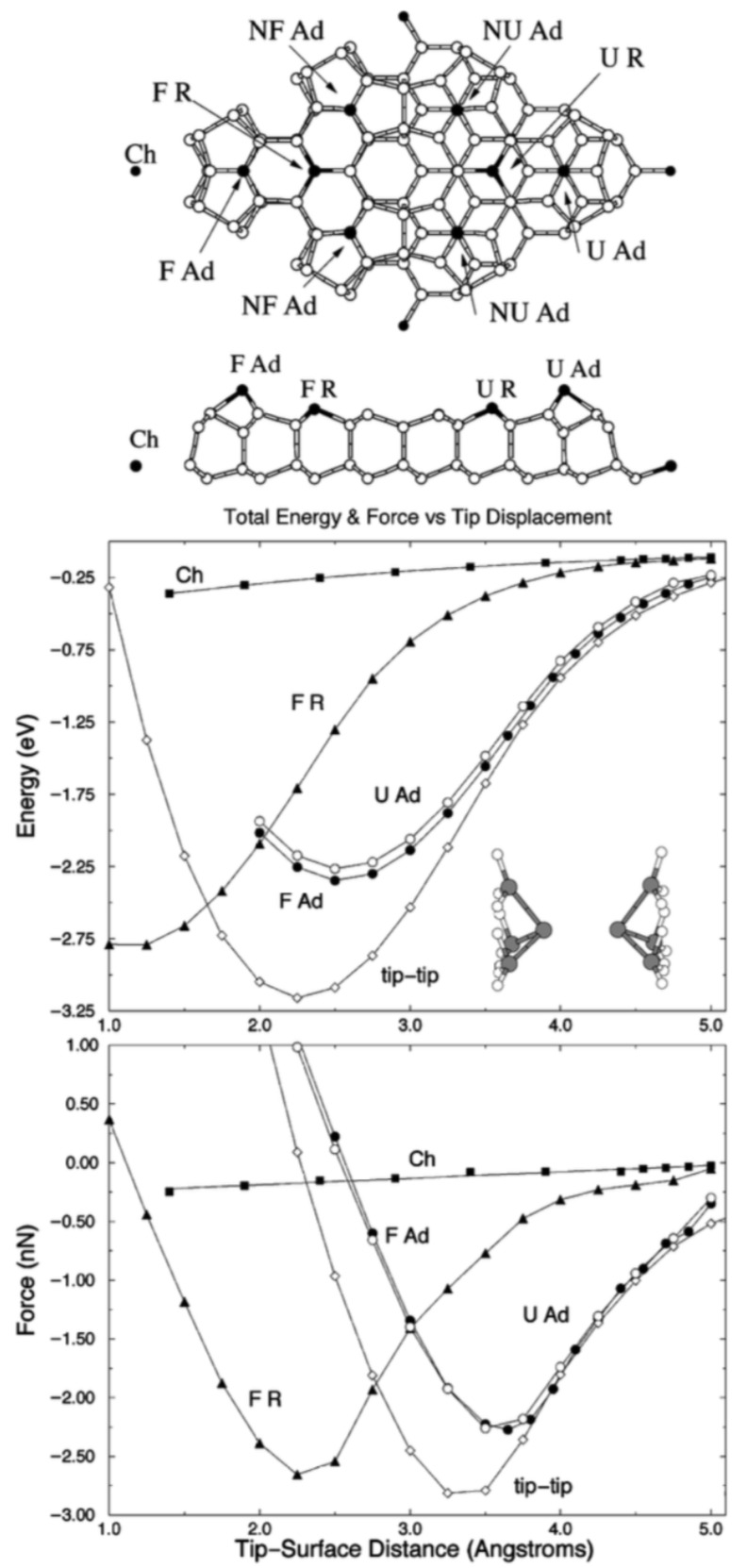
Short-range interaction energy middle panel and normal force bottom panel, as a function of the tip-surface distance, for tip c of Figure 1 over corner hole atom squares, over a rest atom triangles and the adatom on the long diagonal black circles in the faulted half of the unit cell, and over the adatom on the long diagonal in the unfaulted half white circles. For comparison, the interaction between two tips tip b of Figure 1 as shown in the inset, white diamonds is also shown. The top panel shows a ball-and- stick model of the 5 × 5 reconstruction, including a top view of the unit cell and a lateral view of the atoms close to the lattice plane along the long diagonal. The atoms with dangling bonds are marked: corner hole Ch, faulted F R and unfaulted U R rest atoms, faulted diagonal F Ad and off-diagonal NF Adadatoms, and unfaulted diagonal U Ad and off-diagonal NU Ad adatoms. Adapted with permission from American Physical Society from Figure 3 in [36].

**Figure 7 molecules-26-04068-f007:**
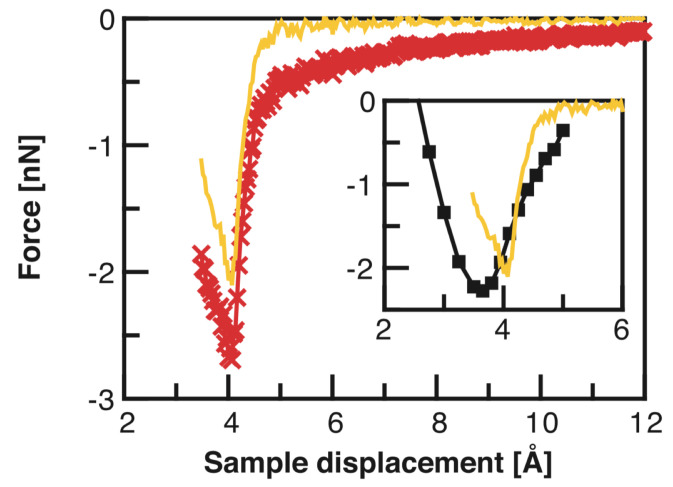
Total force (red line with symbols) and short-range force (yellow line) determined above the adatom site labeled 2 in Figure 1. In the inset, the measured short-range force is compared with a first principles calculation [36] (black line with symbols, see Figure 6). Adapted with permission from American Association for the Advancement of Science from Figure 2C in [38].

**Figure 8 molecules-26-04068-f008:**
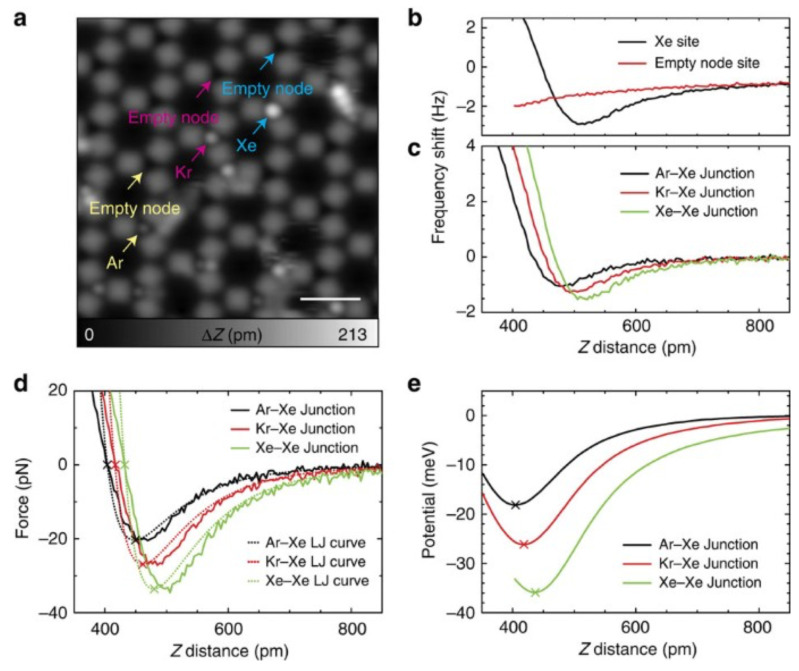
(**a**) STM topography of the Cu-coordinated 3deh-DPDI network after measuring interaction curves above Ar, Kr and Xe atoms. (**b**) Frequency shift curves taken at the equivalent node sites with and without a Xe atom, measured with a Xe-functionalized tip. (**c**) Subtracted distance-dependence curves of the frequency shift for Ar–Xe, Kr–Xe and Xe–Xe junctions. (**d**) Extracted atomic Ar-Xe, Kr-Xe and Xe-Xe interaction forces and Lennard-Jones fits for each system, derived from the frequency shift signal and (**e**) the equivalent potential energy curves. Measurement parameters: $V_{tip}=500$ mV and I=4 pA in a, and A=38 pm, f=23,063 Hz and Q=52,044, and V=1 mV in b,c. Scale bar, 2 nm. Adapted with permission from Springer Nature from Figure 3 in [40].

**Figure 9 molecules-26-04068-f009:**
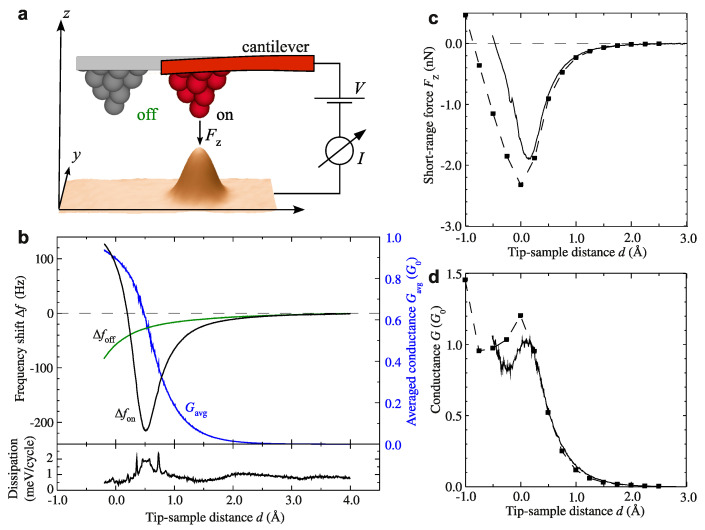
(**a**) 3D representation of a constant current STM image (I=1 nA, V=1 mV, 4×5 nm${}^{2}$) of a single Pt atom adsorbed on a Pt(111) surface and a schematic of the probing tip. (**b**) Time-average junction conductance $G_{av}=I_{av}/V$ (blue dashed curve) and frequency shift Δf on top (black curve) and off (green curve) the Pt atom measured at different tip–sample distances *d*. The lower panel shows the dissipation signal *D* on top of the Pt atom recorded simultaneously with $G_{av}$ and Δf. (**c**) Calculated short-range force between tip apex and Pt adatom (full line) and simulated forces (squares) from data in (**b**). (**d**) Conductance after deconvolution of the tip oscillation (full line) and simulated conductance (squares). All conductances are given in units of the single–channel, spin–degenerate quantum of conductance $G_{0}=2e^{2}/h\approx {\left(12,906\;\Omega \right)}^{-1}$ where *e* is the elementary charge and *h* is Planck’s constant. Adapted with permission from American Physical Society from Figure 1 in [46].

**Figure 10 molecules-26-04068-f010:**
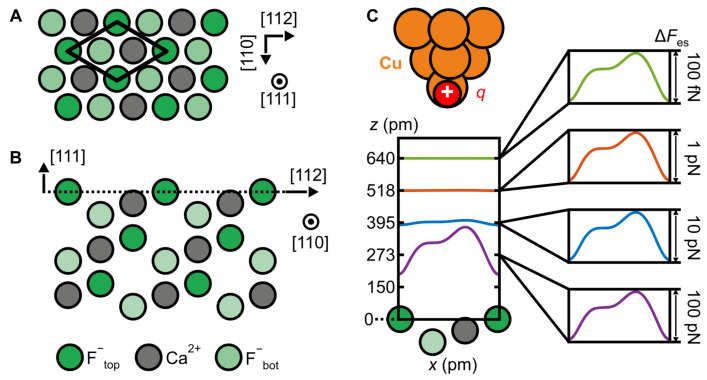
Schematic of the CaF${}_{2}$(111) surface and decay of the short-range electrostatic tip–sample interaction with distance. (**A**) Top view of the CaF${}_{2}$(111) surface. The surface unit cell presents three inequivalent sites, corresponding to the different ions of the topmost triple layer [50]. (**B**) Side view of the CaF${}_{2}$(111) surface. The surface layer is marked by the dashed line. (**C**) A single-atom metal tip has been modeled with a positive point charge *q*. Graph: illustration of the exponential decay of the short-range electrostatic force contrast $\Delta F_{es}$ with tip–sample distance *z* for the single-atom metal tip above the CaF${}_{2}$(111) surface. Adapted with permission from Institute of Physics from Figure 1 in [49].

**Figure 11 molecules-26-04068-f011:**
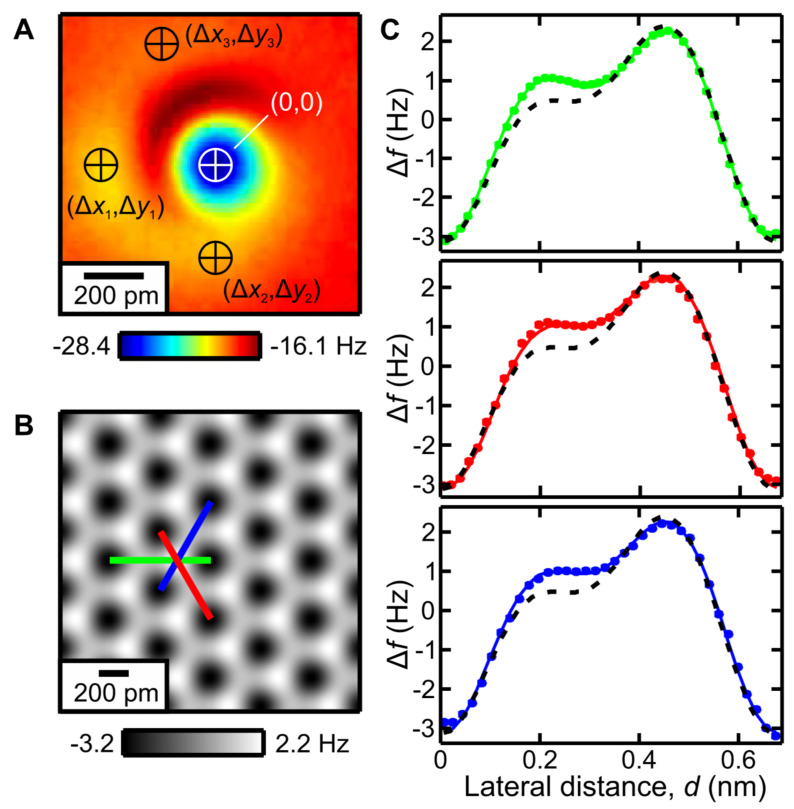
Refined tip apex description based on the COFI characterization. (**A**) COFI image of the single-atom metal tip.The white cross indicates the front atom at coordinates (0,0). Black crosses indicate second layer atoms (shifted with respect to the front atom by ($\Delta x_{i}$, $\Delta y_{i}$) used to refine the electrostatic calculation. Note: yellow color indicates a local minimum. (**B**) Calculated Δf image of CaF${}_{2}$(111) for the multi–atom tip model. (**C**) Line profiles along the three high–symmetry directions marked in (**B**). The calculation including four point charges (solid) matches the asymmetry of the experiments (dots) and gives a better agreement than the single point charge model (dashed). Adapted with permission from Institute of Physics from Figure 3 in [49].

**Figure 12 molecules-26-04068-f012:**
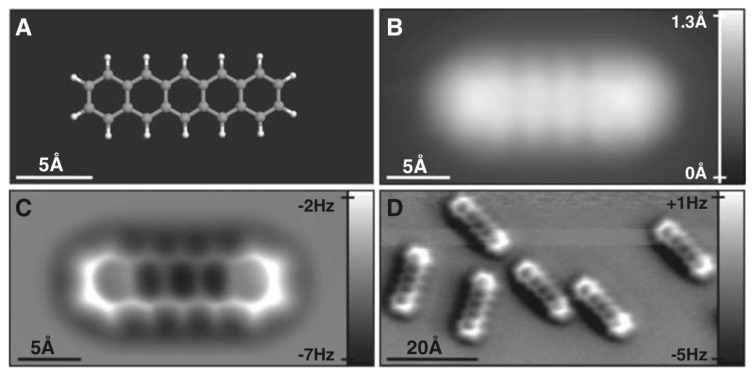
STM and AFM imaging of pentacene on Cu(111). (**A**) Ball-and-stick model of the pentacene molecule. (**B**) Constant-current STM and (**C**,**D**) constant-height AFM images of pentacene acquired with a CO-modified tip. Imaging parameters are as follows: (**B**) set point I=110 pA, V=170 mV; (**C**) tip height z=−0.1 Å (with respect to the STM set point above Cu(111)), oscillation amplitude A=0.2 Å; and (**D**) z=0.0 Å, A=0.8 Å. The asymmetry in the molecular imaging in (**D**) (showing a “shadow” only on the left side of the molecules) is probably caused by asymmetric adsorption geometry of the CO molecule at the tip apex. Adapted with permission by American Association for the Advancement of Science from Figure 1 in [54].

**Figure 13 molecules-26-04068-f013:**
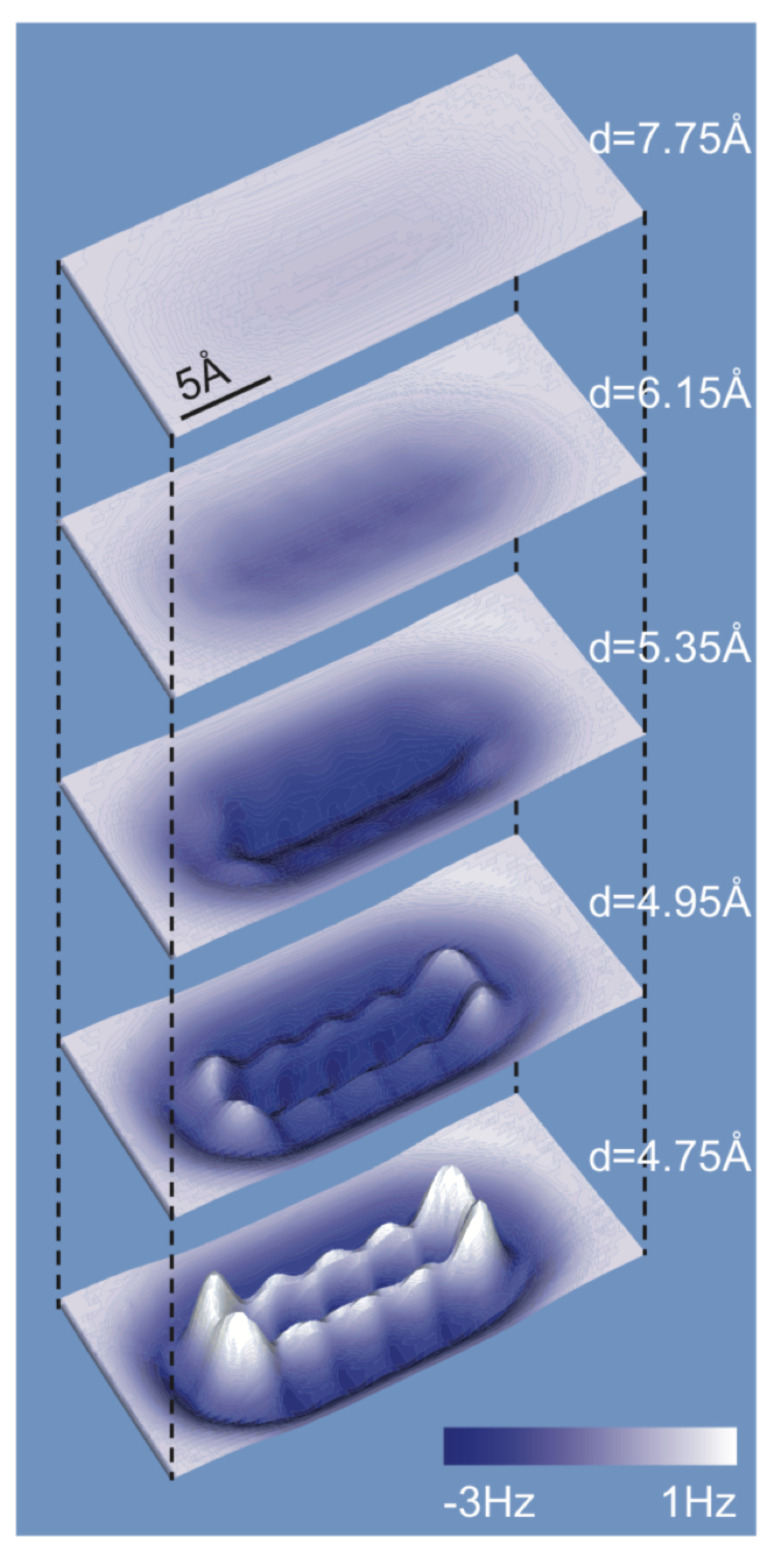
3D contours of computed frequency shift for five different intermolecular distances *d*. The contour area is 23.6 Å × 10.6 Å. Adapted with permission from Institute of Physics from Figure 5 in [64].

**Figure 14 molecules-26-04068-f014:**
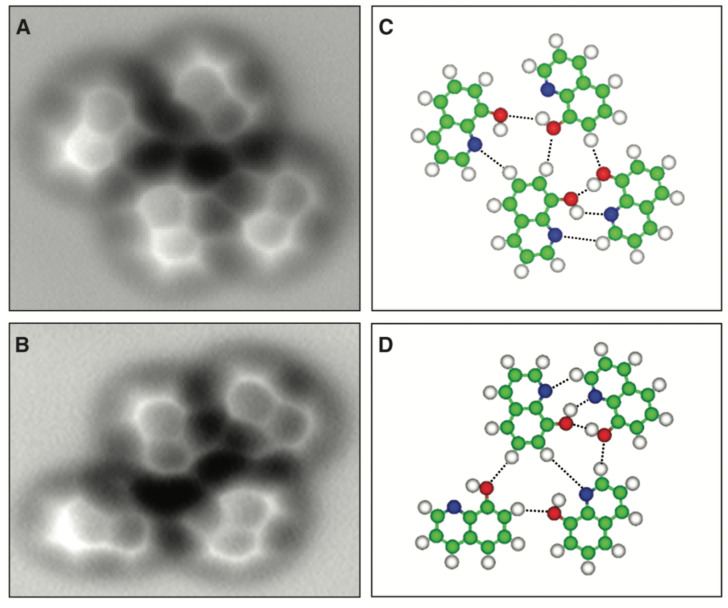
AFM measurements of 8-hydroxyquinoline (hq) molecules assembled clusters on Cu(111). (**A**,**B**) Constant-height frequency shift images of typical molecule-assembled clusters and their corresponding structure models (**C**,**D**). Imaging parameters: V=0 V, A=100 pm, Δz=+10 pm. Image size: (**A**) 2.3 by 2.0 nm; (**B**) 2.5 by 1.8 nm. The dashed lines in (**C**,**D**) indicate likely H bonds between 8-hq molecules. Green, carbon; blue, nitrogen; red, oxygen; white, hydrogen. Adapted with permission from American Association for the Advancement of Science from Figure 2 in [75].

**Figure 15 molecules-26-04068-f015:**
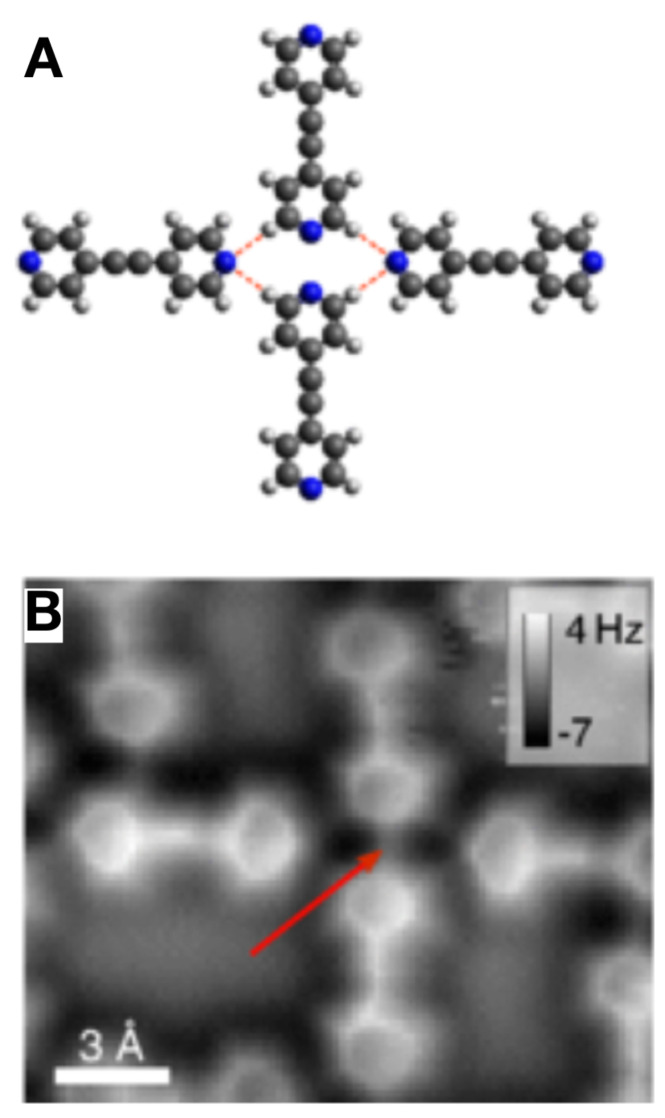
Bis(parar-pyridyl)acetylene (BPPA) molecules. (**A**) Schematic of the tetramer. (**B**) AFM image of the tetramer taken with a CO terminated tip showing apparent intermolecular bonds. Adapted with permission from American Association for the Advancement of Science from Figure 2 in [76].

**Figure 16 molecules-26-04068-f016:**
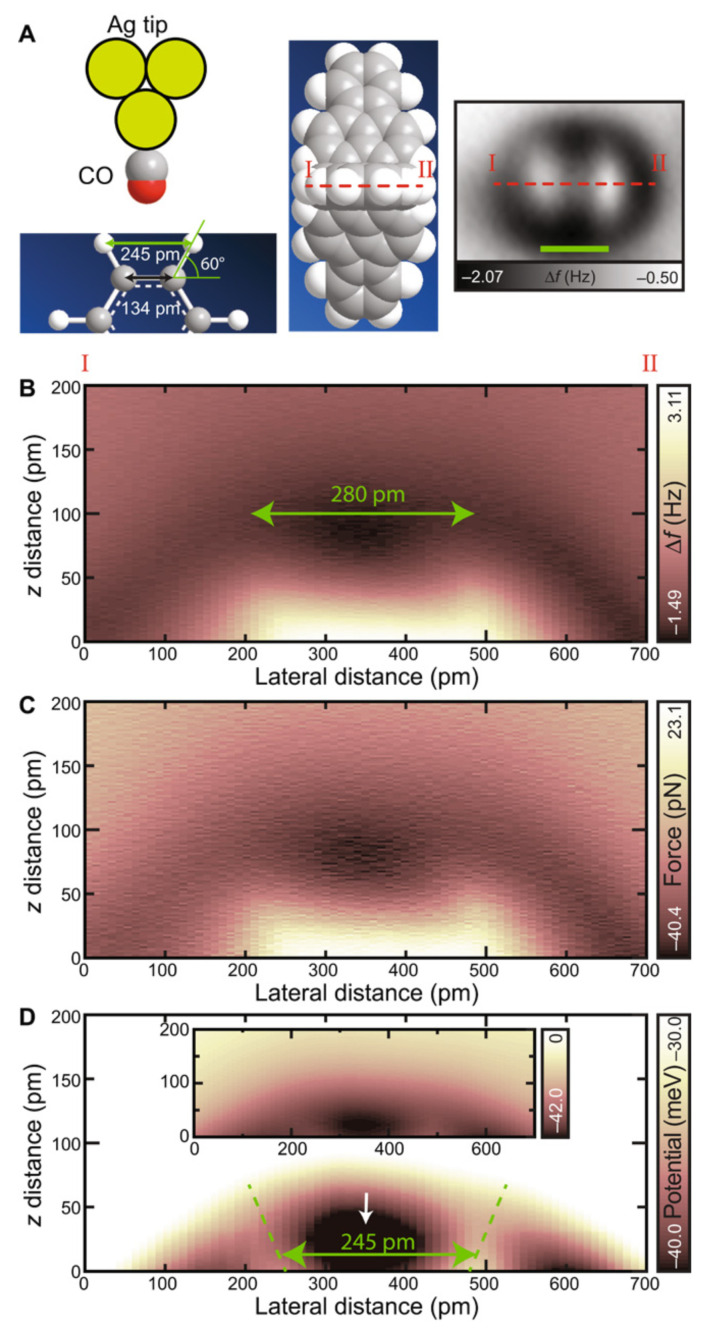
Quantitative measurements of the C-O...H-C bond. (**A**) Schematic drawing of the hydrogen bonding measurement on TFAP with a CO-functionalized tip. Right shows the AFM image. Scale bar, 300 pm. (**B**) Two-dimensional frequency shift map. (**C**) Calculated force and (**D**) potentials. Inset shows the same area with a wider contrast. The z origin was set at the position of the hydrogen atom. Measurement parameters: A=60 pm and V=0 mV. Adapted with permission from American Association for the Advancement of Science from Figure 3 in [79].

**Figure 17 molecules-26-04068-f017:**
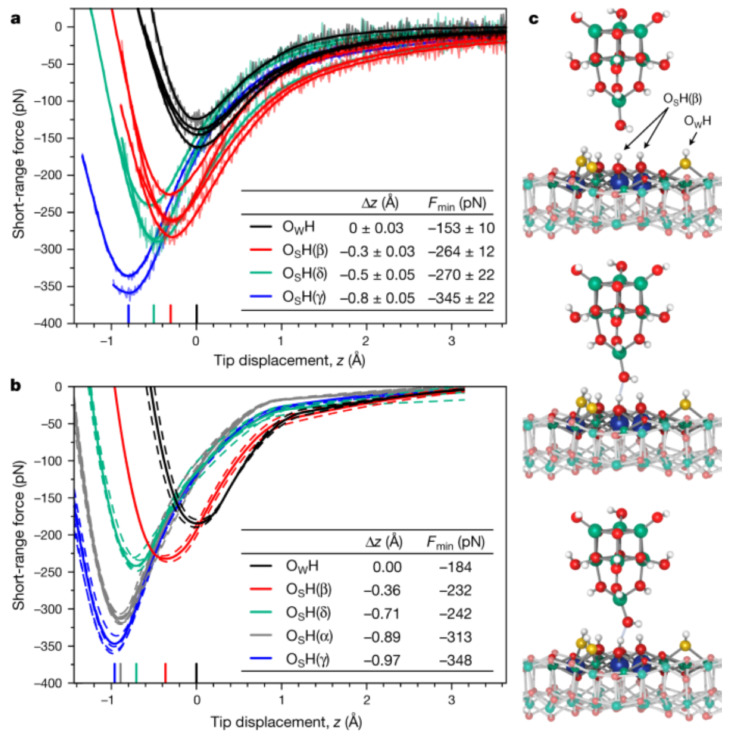
Probing individual surface hydroxyls with the AFM tip. (**a**) Experimental short-range force–distance curves, F(z), for the OH groups initially formed on water dissociation, OWH and OSH(β), as well as for OSH(δ) and OSH(γ) that were constructed by manipulation with the tip. Results from four independent datasets are shown. The tip displacement *z* is given with respect to the position of the OWH force curve minimum (z=0). The positions of the minima are indicated on the *z* axis for all OH groups by short coloured lines; the calculated geometric heights of the O atoms in absence of the tip are listed in Figure 3a of [80]. The errors for the force minimum (Fmin) and displacement (Δz) provided in the inset were derived using Student’s t-distribution with a confidence interval equivalent to 1σ (68.27%). (**b**) Calculated short-range force–distance curves. Dashed curves are individual calculations for different tip orientations and sites in the unit cell (see Methods) and solid lines are the averaged curves. (**c**) Tip–sample configuration for various separations while probing an OSH(β) group. OW, yellow; H, white; O(3c) and O(4c), red; In(6c), blue; In(5c), green. Adapted with permission from Springer Nature from Figure 2 in [80].

**Figure 18 molecules-26-04068-f018:**
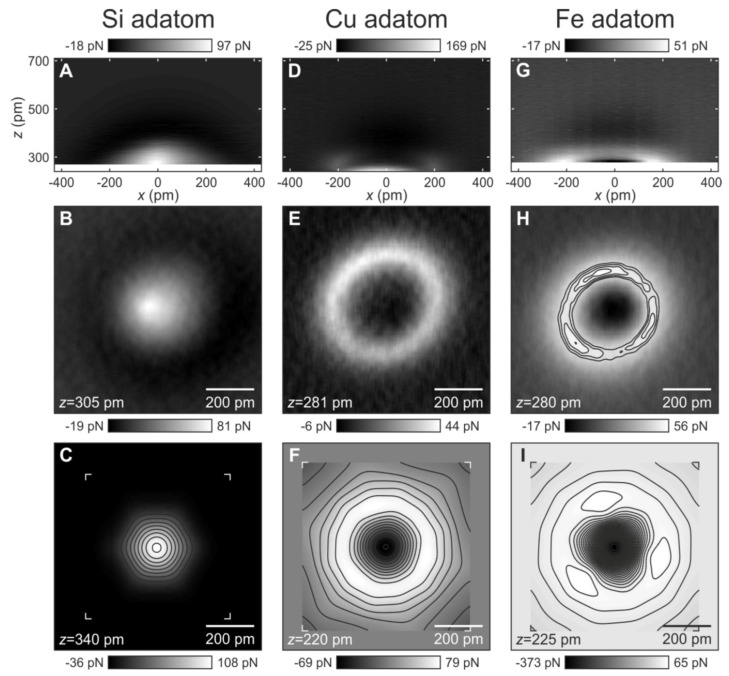
Experimental and calculated forces for three different adatoms in side and top views. Top row (side view): Experimental vertical forces $F_{z}$ in the xz plane between a CO-terminated AFM tip and (**A**) a Si adatom, (**D**) a Cu adatom, and (**G**) a Fe adatom on Cu(111). Middle row (top view): Constant-height force data in the xy plane between a CO-terminated tip and (**B**) a Si adatom, (**E**) a Cu adatom, and (**H**) a Fe adatom on Cu(111) taken at z positions, as indicated in the left bottom corner, respectively. Bottom row (top view): DFT calculations of $F_{z}$ in the xy plane between a CO molecule tip and (**C**) a Si adatom, (**F**) a Cu adatom, and (**I**) a Fe adatom on Cu(111). The three local maxima on the experimental data (**H**) and DFT data (**I**) for the Fe adatom are located above the hollow sites of the Cu(111) substrate underneath (see Figures S5 and S6 in [71]). Note that the color scale is the same for the force data in the top and middle rows. The color scale in the bottom row is different in order to maximize the contrast. Scale bars, 200 pm. Adapted with permission by American Association for the Advancement from Figure S11 in [71].

**Figure 19 molecules-26-04068-f019:**
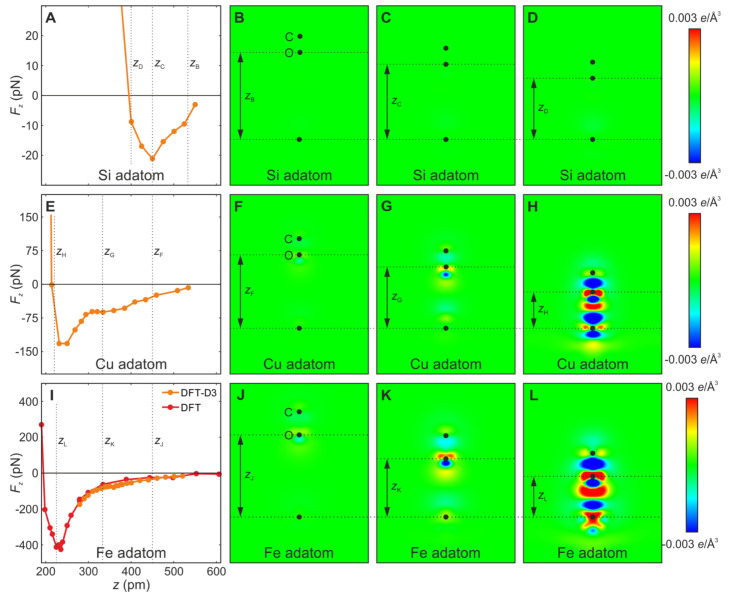
DFT and DFT-D3 calculations of vertical force versus tip-sample distance curves (left column) and of differential charge densities for a (**A**–**D**) Si adatom, (**E**–**H**) Cu adatom and (**I**–**L**) Fe adatom interacting with a CO-terminated tip for three different tip-sample distances as indicated in (**A**), (**E**) and (**I**), respectively. (**A**) For the Si adatom, only vdW attraction occurs with merely 30 pN attraction before Pauli repulsion occurs. (**B**–**D**) The charge density plots show that no significant change in charge density occurs. (**E**) For the Cu adatom, we see, in contrast to the Si adatom, two local force minima: a shallow one due to physisorption and a deeper one with a maximal attraction of −132 pN due to chemisorption. (**F**–**H**) The differential charge density plots, in particular the one at the closest distance zH shown in (**H**), reveal significant charge redistribution due to hybridization, pointing to the formation of a chemical bond. (**I**) For the Fe adatom, we find a very shallow local vdW minimum caused by physisorption followed by a global force minimum of −426 pN due to chemisorption. (**J**–**L**) The differential charge density plots already show significant hybridization at a distance $z_{K}$ and more so at $z_{L}$, pointing to the formation of an even stronger chemical bond compared to the case of the Cu adatom. Adapted with permission by American Association for the Advancement of Science from Figure S7 in [71].

**Figure 20 molecules-26-04068-f020:**
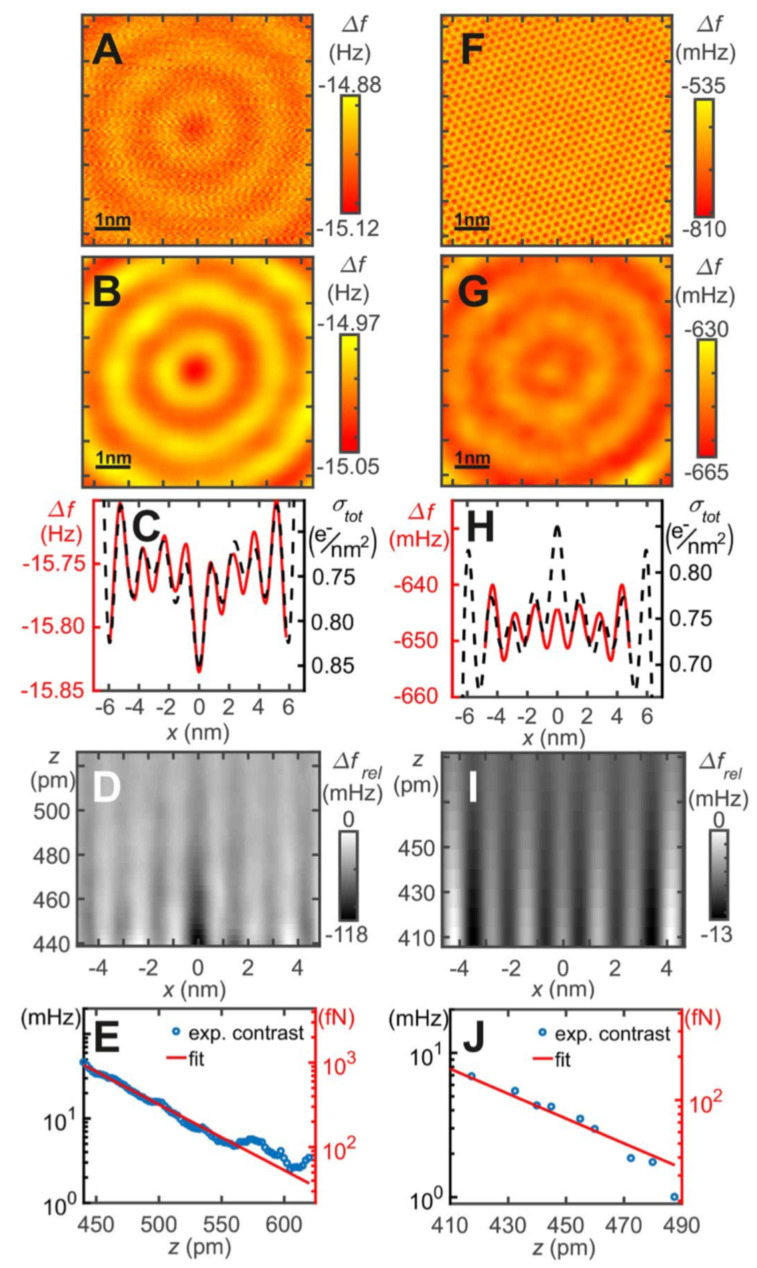
AFM data inside the quantum corral for a single atom metal tip (left column) and a CO terminated tip (right column). (**A**) Frequency shift data Δf(x,y,z=const.) with an attractive interaction between the probe tip and the Bessel–type eigenstates. (**B**) Low–pass filtered version of A. (**C**) Profile line through the center of B and reversed total surface charge density of the corral. (**D**) Gray scale representation of offset and slope corrected frequency shift $\Delta f_{rel}\left(x,0,z\right)$. (**E**) Distance dependence of the contrast, defined as the difference between the average of the first and second maxima and the minimum in between. (**F**) Frequency shift data Δf(x,y,z=const.), showing attraction to the Cu surface atoms and repulsion to the Bessel-type eigenstates. (**G**) 10 Low-pass filtered version of F, showing only the repulsion to the Bessel-type eigenstates. (**H**) Profile line through center of G compared to total charge density. (**I**) Gray scale presentation of offset and slope corrected frequency shift $\Delta f_{rel}\left(x,0,z\right)$. (**J**) Contrast, defined by the difference between the first maximum at x≈±1.3 nm and the average between the first and second minimum, evolution as a function of vertical distance *z*. See section 7 of [84] for details of data extraction. Adapted with permission by American Association for the Advancement of Science from Figure 2 in [84].

**Figure 21 molecules-26-04068-f021:**
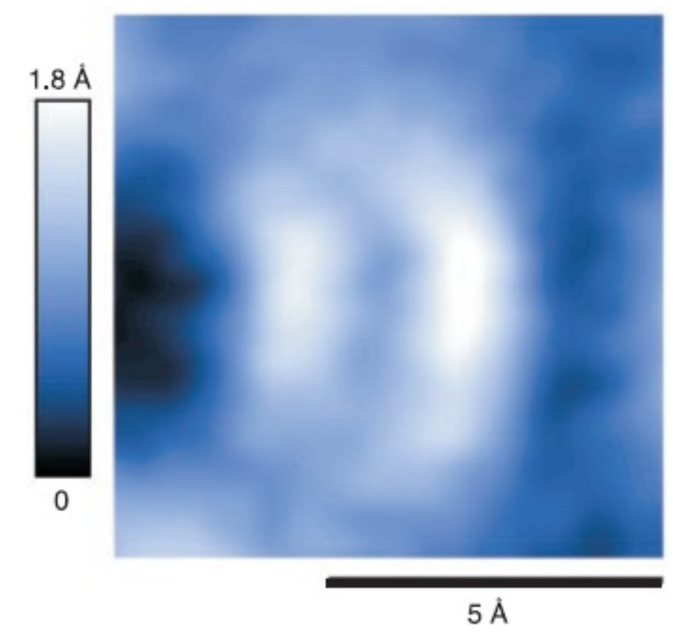
Topographic AFM data of a single adatom in Si(111)–(7×7) recorded with a room temperature UHV AFM fitted with a qPlus sensor with a tungsten tip, presumably terminated by a silicon cluster that points in a [001] direction exposing two dangling bonds. Adapted with permission by American Association for the Advancement of Science from Figure 3 in [17].

**Figure 22 molecules-26-04068-f022:**
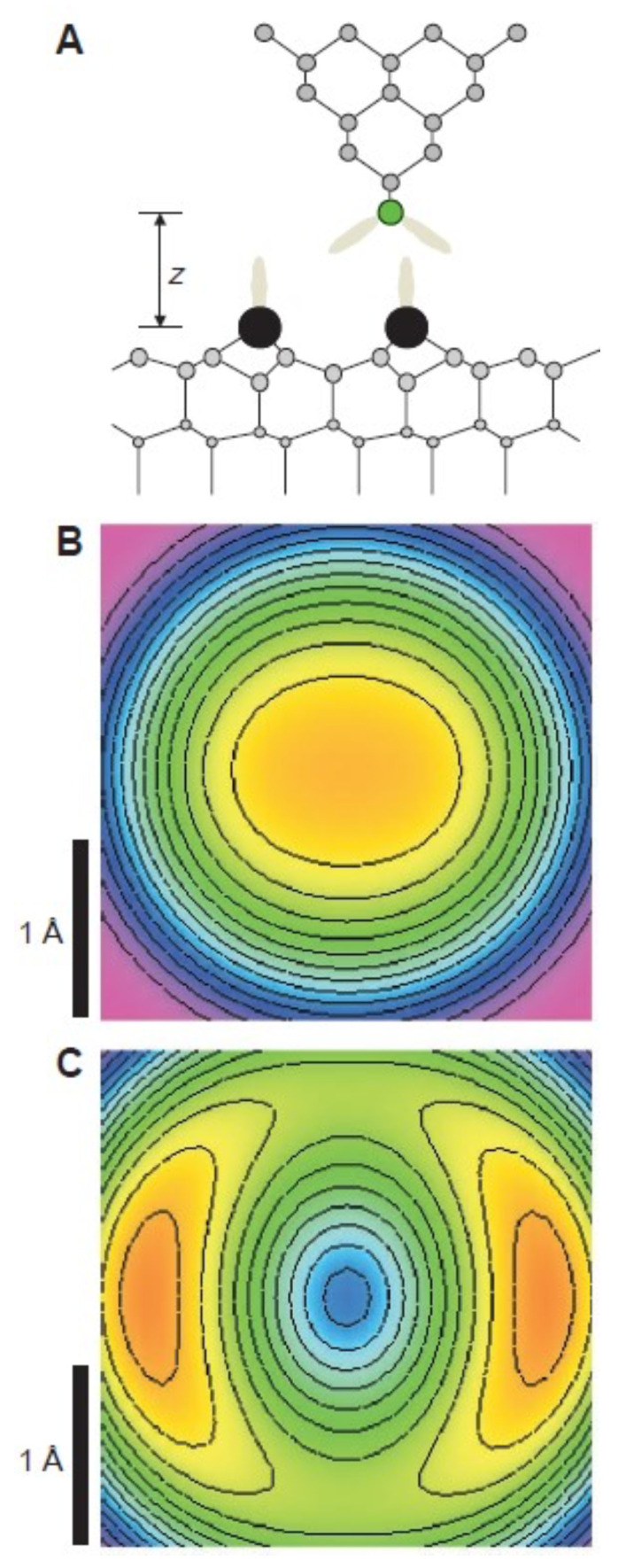
(**A**) Schematic drawing of the geometry used for the calculation, sketching the 3sp${}^{3}$ hybrid orbitals of the tip atom (green sphere) and sample adatoms (black spheres). Simulated 2.8 Å by 2.8 Å constant height images of a single adatom (**B**) at z=310 pm and (**C**) z=285 pm. Adapted with permission by American Association for the Advancement of Science from Figure 4 in [17].

**Figure 23 molecules-26-04068-f023:**
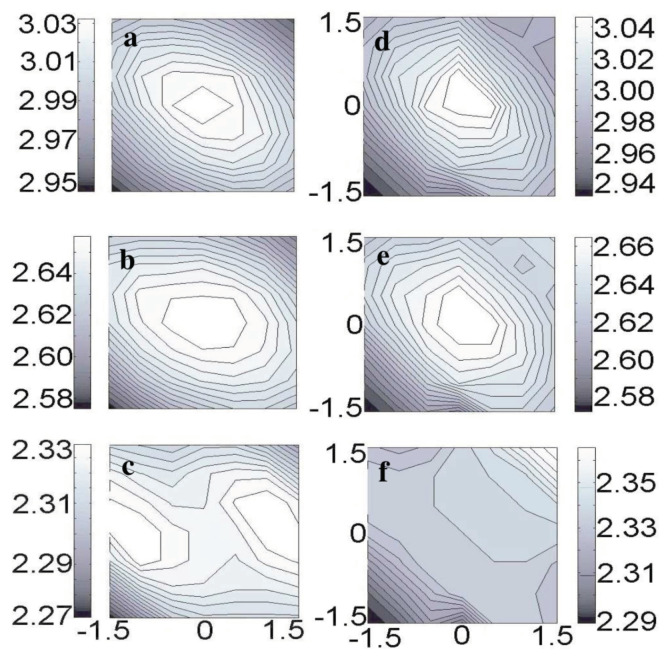
Simulated topographical FM-AFM images at three different frequency shifts (Δf), calculated using $f_{0}=16860\;$Hz, k=1800 N/m, and A=8 Å [17]. Height variations (Å) are indicated by the bar next to image. The left panel is for a Si(001) tip: (**a**) at Δf=−140 Hz, a single maximum; (**b**) at Δf=−150 Hz, still a single maximum but with extended maximum area; (**c**) at Δf=−160 Hz, two maxima separated by 2 Å. The right panel is for Si(111) tip: at all frequency shifts, only one maximum with its position shifted from the center (**d**), (**e**) to the upper-right corner (**f**). Adapted with permission by American Physical Society from Figure 4 in [86].

**Figure 24 molecules-26-04068-f024:**
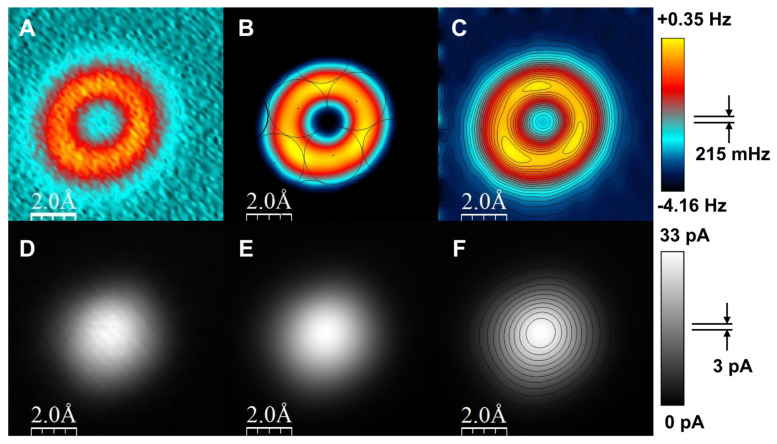
Experimental constant-height images of a Fe adatom on Cu(111) imaged with a CO-terminated tip. The data directly shows the experimental signal, i.e., the frequency shift rather than the forces that are computed from the frequency shift and shown in Figure 2 and Figure S4 from [71]. (**A**–**C**) shows the frequency shift channel in the top row and the (**D**–**F**) simultaneously recorded current channel in the bottom row. For both rows, the left column shows raw data, the images in the central and right column are filtered (re-dimensioned to 512×512 pixels and Gauss-filtered with a 32 pixels width). Parameters were qPlus sensor with $f_{0}=20,430$ Hz, k=1800 N/m, A=25 pm, Q=182,750; $V_{bias}=100\mu$V, acquisition time 50 s, image size 1 nm${}^{2}$ and 64 × 64 pixels, scanning speed 2.56 nm/s. The black circles in (**B**) indicate the surface layer of the Cu(111) plane. The Fe adatom sits on an fcc site, the three local maxima in the repulsive ring are located over hollow sites in agreement with DFT calculations. Vertical scale ranges from −3 Hz to +0.35 Hz. In (**C**), the same data as in (**B**) is shown with contour lines and a slightly changed vertical scale ranging from −4.16 Hz to +0.35 Hz. We note that not all CO-terminated tips allow getting close enough to resolve the three local maxima on the repulsive ring. From our experience, it is best to use relatively sharp tips that show a small vdW background. We speculate that for blunt tips, the vdW interaction between the adatom and the tip is so large that these vdW forces lead to lateral manipulation of the Fe adatom at the close distance that is required to resolve the three local maxima. While we have observed Fe adatom data that shows these three bumps with about a dozen different tips, the data shown here is of a high symmetry, showing that the CO-terminated tip was nearly perfectly vertical (for a different case, see Figure S6 in [71]). Adapted with permission by American Association for the Advancement of Science from Figure S5 in [71].

## Data Availability

For data availablity, please consult the source publications of this review.

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
