# Peer review of "Probing the Nature of Chemical Bonds by Atomic Force Microscopy"

_molecules, 2021, doi:10.3390/molecules26134068_

Round 1

Reviewer 1 Report

This is a very interesting and excellently written manuscript. I very strongly support publication.

Two smaller points:

First of all, when results from other methods than AFM are presented, this should be clearly stated. As an example, figure 4 is based on DFT results but this in not mentioned in the figure caption.

Secondly, the interaction energies and interaction energy curves for noble atom (Xe, Ar) interactions are presented in Figure 7. However, it is not clear how the energy results have been obtained? Is this by integrating the force curve, by integrating and assuming a potential energy expression (function), or purely by calculation? The method used needs to be declared and explained.

Author Response

Dear Reviewer 1,

thank you very much for your helpful comments. Please find the reply below:

First of all, when results from other methods than AFM are presented, this should be clearly stated. As an example, figure 4 is based on DFT results but this in not mentioned in the figure caption.

Thank you very much for this comment. We now state clearly in the figure caption and in the text that this data is based on DFT. In addition, we added an experimental AFM figure before this figure because it is, as you implicitely pointed out, odd to start the results of a review about an experimental study with calculated images.

Secondly, the interaction energies and interaction energy curves for noble atom (Xe, Ar) interactions are presented in Figure 7. However, it is not clear how the energy results have been obtained? Is this by integrating the force curve, by integrating and assuming a potential energy expression (function), or purely by calculation? The method used needs to be declared and explained.

Thanks again, we have elaborated this by adding:

---------------

Taking the difference of force spectra aquired over an empty node in the framework and over a node filled with a noble gas atom allowed to extract the atom-atom interactions as illustrated in figures \ref{fig7}a,b.

Kawai et al. found an interaction energy of 18, 26 and 36\,meV for Xe-Ar, Xe-Kr and Xe-Xe junctions, respectively, as displayed in figure \ref{fig7}. The energy curves were integrated from the experimental frequency shift data and the atom-atom interactions were extracted by subtracting the background forces as explained in figures \ref{fig7}a,b and \cite{Kawai2016}.

----------

Reviewer 2 Report

The paper is an extensive review regarding AFM characterization. The author reported a decent review of the literature scope of AFM measurement of various atomic bonding such as an ionic, covalent, and hydrogen. However, the abstract looks misleading as if it is a research article. Also, I am wondering “qPlus forsce sensor” mentioned in the abstract was solely used in literatures reviewed in this review? Section should be accordingly addressed instead of “method” and “result”. Aside from this, the manuscript is worthy of publication in the journal.

  1. Typo: Figure numbering “Figure 1” in line 86 ->”figure 2”  / “Figure2” in line 100 - > “figure 3”
  2. Typo: “embodment” in line 97
  3. Figure3 could be redrawn since it is hard to understand.

Author Response

Dear Reviewer 2,

thank you very much for your helpful critique and for doing it so quickly.

Your critique about the abstract is very valid, in response the revised abstract clearly says that the paper is a review, not a research article.

Also, you wonder whether the review only considers experiments that were done with qPlus sensors. This is not the case, even some figures have been obtained by using traditional Si cantilevers. This is now stated very clearly.

Thank you for pointing out the typos in 1. and 2. - they have all been corrected.

Figure 3 has been supplemented with a more detailed description in the figure caption and specific referals to photographs of the sensor in the literature that provide detailed information.